# TGLFusion: A Temperature-Guided Lightweight Fusion Method for Infrared and Visible Images

**DOI:** 10.3390/s24061735

**Published:** 2024-03-07

**Authors:** Bao Yan, Longjie Zhao, Kehua Miao, Song Wang, Qinghua Li, Delin Luo

**Affiliations:** 1School of Aerospace Engineering, Xiamen University, Xiamen 361102, China; 23220211151692@stu.xmu.edu.cn (B.Y.); lzha0538@uni.sydney.edu.au (L.Z.); 23220211151662@stu.xmu.edu.cn (Q.L.); luodelin1204@xmu.edu.cn (D.L.); 2Electric Power Research Institute, China Southern Power Grid, Guangzhou 510063, China; wangsong@csg.cn

**Keywords:** deep learning, image fusion, infrared and visible sensor images, lightweight model, electric power equipment

## Abstract

The fusion of infrared and visible images is a well-researched task in computer vision. These fusion methods create fused images replacing the manual observation of single sensor image, often deployed on edge devices for real-time processing. However, there is an issue of information imbalance between infrared and visible images. Existing methods often fail to emphasize temperature and edge texture information, potentially leading to misinterpretations. Moreover, these methods are computationally complex, and challenging for edge device adaptation. This paper proposes a method that calculates the distribution proportion of infrared pixel values, allocating fusion weights to adaptively highlight key information. It introduces a weight allocation mechanism and MobileBlock with a multispectral information complementary module, innovations which strengthened the model’s fusion capabilities, made it more lightweight, and ensured information compensation. Training involves a temperature-color-perception loss function, enabling adaptive weight allocation based on image pair information. Experimental results show superiority over mainstream fusion methods, particularly in the electric power equipment scene and publicly available datasets.

## 1. Introduction

The fusion of infrared and visible images is crucial in computer vision. Real-world applications are often influenced by factors such as lighting and weather conditions. Using visible or infrared images alone may struggle to meet the diverse requirements of different environments. Combining these two spectral images provides a more comprehensive scene understanding, aiding in improving the accuracy of various computer vision tasks such as object detection, recognition [1], and tracking [2]. In practical applications, image fusion is also employed in fields like medical diagnostics, surveillance systems, military reconnaissance, and providing essential support for a variety of computer vision tasks in different scenarios [3].

In recent decades, researchers have developed traditional image fusion methods [4], utilizing mathematical transformations to convert source images into the transform domain for measuring activity levels.

In recent years, the rapid development of deep learning has stimulated exploration in data-driven approaches for image fusion within the field [5]. Based on the utilized benchmarks, mainstream data-driven methods can be roughly categorized into four types: auto-encoder (AE)-based methods, convolutional neural network (CNN)-based methods, generative adversarial network (GAN)-based methods, and transformer-based methods.

There are notable differences between power equipment scenes and daily life scene characteristics. Power equipment scenes typically include substations, power transmission lines, and power towers, which generate significant amounts of heat, resulting in prominent thermal features in infrared images. Visible images, on the other hand, can reveal the appearance and structure of the power equipment. In contrast, daily life scenes encompass common objects, such as people, buildings, and natural landscapes, where the thermal features are relatively less prominent. The scene information is primarily captured through visible images with existing research mainly focusing on everyday scenes such as transportation and roads [6], with limited and relatively weak studies dedicated to power equipment scenes. In this study, the proposed fusion method can effectively address power equipment scenes with prominent thermal features.

In summary, the primary contributions of this study can be summarized as follows:Based on a comparative study focusing on power equipment and other heat-emitting scenarios, the pivotal role of temperature information in the fusion process of these scenes was confirmed. With this understanding, TGLFusion introduced a temperature-aware optimization weight allocation module tailored specifically for infrared images. This module calculates multispectral weights using a temperature distribution mechanism based on high-temperature ratios, aiming to represent the contributions of source images during the fusion process. Multispectral weights are adaptively assigned to more effectively fused image information. Guided by the temperature loss function, this model optimizes and integrates fusion images based on the thermal information from infrared images, significantly increasing the information content in the fused images.The backbone network of this model is composed of the MobileBlock framework and the MICM (Multispectral Information Complementary Module). During the feature extraction process, a feature-enhancement attention mechanism extracts and enhances unique features in various spectra. This approach effectively reduces redundant information while preserving complementary information.Through objective and subjective experiments, TGLFusion was compared with six mainstream fusion models, demonstrating significant advantages in the evaluation metrics for power equipment image fusion. This validates the importance of our model in the field of power equipment image fusion.

These contributions collectively enhance the understanding and effectiveness of image fusion in the context of power equipment, offering valuable insights and a practical model for improving the image quality and information extraction in this domain.

## 2. Related Work

In recent years, significant progress has been made in methods that utilize deep learning to construct fusion network models for infrared and visible image fusion [7]. These methods can be broadly classified into four categories: CNN-based methods, autoencoder-based methods, GAN-based methods, and transformer-based methods.

### 2.1. CNN-Based Methods

Based on our understanding, Liu et al. [8] were among the pioneers of CNN-based methods for visible and infrared fusion. In their research, they employed a Siamese CNN to generate a set of weight maps and conducted Laplacian pyramid decomposition on source images, along with Gaussian pyramid decomposition on the weight maps, subsequently conducting fusion using a multi-level approach. The training of this model utilized high-quality images alongside their blurred counterparts, which were created using multiscale Gaussian filtering and random sampling.

Many unsupervised CNN methods involve the selective utilization of CNN in particular stages of the process. For instance, Liu et al. [9] segregated source images into fundamental and intricate elements. They subsequently merged the fundamental elements employing a weighted summation technique, while the intricate components were fused using a CNN alongside a multi-layer feature fusion approach. Hou et al. [10] proposed the VIF-Net, utilizing CNN for feature extraction and image reconstruction. There are also unsupervised CNN methods where CNN is applied throughout the entire process. For example, Xu et al. [11] and Mustafa et al. [12] used CNN in all three stages.

Multiple strategies have been applied to enhance the performance of CNN-based methods. In 2019, Li et al. introduced residual connections [13], marking a significant breakthrough in the field. This technique was subsequently adopted by some famous methods [14], demonstrating a remarkable performance. Another widely adopted technique for performance enhancement is the integration of attention mechanisms. Various types of attention mechanisms have been employed, such as channel attention [15] and spatial attention [16]. Additionally, dense connections, an important innovation introduced by Li et al. in 2018 [17], were used to enhance representational capabilities within the encoder. Since then, dense connections have been extensively employed in various methods, significantly impacting performance improvements.

### 2.2. Autoencoder-Based Methods

DenseFuse [17] stands out as a groundbreaking autoencoder (AE)-based fusion method. This technique involves pretraining the AE and employing diverse fusion strategies for feature integration. Similarly, Raza et al. [18] proposed an AE-based fusion method utilizing a weight map-driven fusion strategy derived through softmax operations based on extracted features. Additionally, Fu et al. [19] introduced an AE-based approach featuring dual branches within the encoder.

AE-based methods often train the autoencoder exclusively using RGB images and then directly apply the trained autoencoder to infrared images [20]. However, this approach might encounter performance limitations due to the inherent disparities between RGB and infrared images. To address this challenge, a certain method [21] adopts an alternating training approach that utilizes both RGB and infrared images to train the autoencoder.

### 2.3. GAN-Based Methods

In 2019, Ma et al. [22] pioneered the integration of generative adversarial networks (GANs) into the field of image fusion. This led to a series of GAN-based fusion methods emerging, establishing this category of methods as crucial and influential within the domain.

Several strategies have been employed in these GAN-based methods to enhance fusion performance. For instance, Xu et al. [23] introduced the local binary pattern loss during training; Xu et al. [24] incorporated residual blocks and skip connections into the generator architecture, while Fu et al. [25] proposed using dense blocks to enhance the generator’s information capture capability.

However, these GAN-based methods commonly share a limitation: typically solely utilizing a single discriminator, which tends to bias the generated fused image toward resembling either visible images [26] or infrared images. In such cases, as adversarial training progresses, the fused image may lose certain details from the source images. To address this issue, Ma et al. [27] suggested the utilization of a multi-classification-based discriminator to establish equilibrium between the distribution of visible and infrared components.

### 2.4. Transformer-Based Methods

In 2021, transformers made their debut in the realm of image fusion, heralding a series of transformer-based approaches tailored for visible and infrared fusion (VIF) [28] as well as general image fusion.

These approaches selectively employed transformers for feature fusion. For instance, Zhao et al. [28] introduced the DNDT method, employing dual transformers as the fusion strategy. VS et al. [29] promoted a multi-scale fusion approach employing transformers to merge local and global information simultaneously. The method was devised with an encoder for feature extraction from source images and a decoder to produce the fused image. Liu et al. [30] developed a transformer fusion block that employs focal self-attention to combine features derived from a multi-scale encoder. The ultimate fused image is produced via a decoder that incorporates nested connections.

More recently, Rao et al. [31] proposed a visible and infrared fusion (VIF) method combining transformers with generative adversarial networks (GANs). In this method, the generator integrates both spatial and channel transformers to form a transformer fusion module. Additionally, Ma et al. introduced SwinFusion [32], a general image fusion method based on the Swin transformer. They elucidated the critical role of global information in image fusion and provided visualizations of its impact. To our knowledge, SwinFusion represents the first explicit study within the image fusion field that emphasizes the significance of global information.

## 3. Proposed Method

### 3.1. Temperature-Guided Mechanism Module

Given the infrared image $I_{ir}$ and visible image $I_{vi}$, the fused image $I_{f}$ can be generated through feature extraction, fusion, and reconstruction. To improve the fusion performance, TGLFusion introduced a newly designed loss function that considers the temperature information of the infrared image to constrain these three steps.

The areas with higher temperatures indicate more severe heating conditions, which should be preserved and highlighted in the fused image. For infrared images, there is a direct proportional relationship between pixel values and temperature [33], and regions with high pixel values indicate areas with high temperatures.

Considering that temperature imbalance affects the distribution of information, a temperature-guided mechanism module has been developed to estimate the high-temperature proportion of the infrared image. Given an infrared image $I_{ir}$, the high-temperature proportion is defined in (Equation 1).
(1)$$r=\frac{count(P_{i}\geq T)}{N},1\leq i\leq N$$
where count(·) is used to count the number of elements that satisfy the specified condition. $P_{i}$ represents the *i*-th pixel in the entire infrared image. *T* is the high-temperature threshold, and *N* represents the total number of pixels in the image. The high-temperature ratio refers to the ratio between the number of pixels in an infrared image with grayscale values greater than or equal to *T* and the total number of pixels in the entire image.

The temperature information reflects the richness of details in the infrared image, as the high-temperature areas contain more meaningful information. Hence, through a temperature-aware weight distribution mechanism, the high-temperature proportion is utilized to calculate the multispectral weights representing the source contributions. To simplify the computation, our weight allocation mechanism adopts the following straightforward function definitions in (Equation 2) and (Equation 3).
(2)$$W_{ir}=\frac{1}{1+e^{-\alpha (r-\beta })}$$
(3)$$W_{vi}=1-W_{ir}$$
where $W_{ir}$ and $W_{vi}$ represent the contributions of the infrared image and visible image to the fused image, respectively. α is an amplification factor to enlarge the infrared contribution, and β is a threshold to calibrate whether the high-temperature proportion is high or low.

In this paper, the testing conducted on the infrared images of the MSRS dataset [6] revealed that images with richer infrared information have calculated *r* greater than 30, while images with less infrared information have calculated an *r* less than 30. Therefore, β was set to 30 as a criterion to measure the richness of infrared information, indicating that, if the high-temperature regions occupy 30% of the total infrared image, the infrared is considered to contain more meaningful information. Thus, the contribution of the infrared is increased based on (Equation 2). Conversely, if the high temperature areas constitute less than 30%, the visible spectrum will make a larger contribution.

### 3.2. Model Structure

The forward propagation pathway of the model consists of a feature extraction block and a feature reconstruction block in Figure 1 and Table 1.

In the feature extraction block, encoders are set up to extract high-level representations from the visible image and infrared image, respectively. Each encoder consists of two MobileBlock convolutional structure blocks and MICM. Given the infrared image $I_{ir}$ and visible image $I_{vi}$, the encoders extracted high-level infrared feature $F_{ir}$ and visible feature $F_{vi}$, and then passes through two 1 × 1 convolution layers and two fully connected layers to obtain global features $G_{ir}$ and $G_{vi}$. Finally, the channel concatenation is utilized to integrate $F_{ir}\cdot G_{ir}$ and $F_{vi}\cdot G_{vi}$.

In the feature reconstruction block, a decoder is set up to fuse and reconstruct the image from the concatenated high-level visible and infrared features. The decoder consists of two MobileBlock convolutional blocks linked by a 1 × 1 convolutional block. After convolution through these three structures, the concatenated features extracted by the encoder are transformed into an image with the same channels and dimensions as the input images. This is the final fused image obtained through the forward propagation pathway.

Along with the input visible and infrared images, the edge contour texture information $E_{ir}$ and $E_{vi}$ are extracted from them using Canny edge detection and threshold filtering, respectively. These data are utilized to fit the edge texture loss function during backpropagation. Furthermore, the high-temperature pixel information is obtained from the infrared image via threshold segmentation. This is leveraged to calculate the pixel weight values $W_{ir}$ and $W_{vi}$ for the infrared and visible images. These weights are applied when fitting the temperature color loss function during backpropagation.

MobileBlock and MICM are both implemented using PyTorch and integrated into TGLFusion. In terms of model lightweighting, MobileBlock significantly reduces the model’s footprint and computational load by adopting a lightweight convolutional structure. Regarding information compensation, MICM obtains spectral difference features by subtracting one spectral feature from another, amplifies and highlights the difference features, and adds them back to the original spectral features. At this point, both spectral features include information compensation for each other, ensuring the mutual complementarity of information in the subsequent fusion process.

### 3.3. MobileBlock

In this section, a convolutional block structure named MobileBlock for the image fusion model is designed, as shown in Figure 2 and Table 2.

The MobileBlock structure consists of five modules: dilated convolution layer, depthwise separable convolution layer, mapping convolution layer, residual connection, and SE layer [34]. The first module is the dilated convolution layer. It expands the number of channels of the input feature map by a dilation factor *t* through a 1 × 1 convolution kernel. This introduces more channel information with a relatively low computational cost. In the depthwise separable convolution layer, each input channel is convolved with an independent 3 × 3 kernel, generating an intermediate feature map with the same number of channels. This allows feature extraction from each channel without additional computation. Next, the mapping convolution layer performs a 1 × 1 convolution on the intermediate feature map to map the channels to the desired output channels. This is equivalent to linearly combining each pixel across channels to fuse information. The residual connection adds the input and the output of the three convolution layers through a 1 × 1 convolution. This facilitates feature reuse and strengthens representational capacity.

### 3.4. MICM

Additionally, as shown in Figure 3, a multispectral information complementary module (MICM) was designed to amplify the differences between infrared and visible images. The difference information is then, respectively, added to their own features to achieve the complementary module. The multispectral information complementary module can be defined in (Equation 4) and (Equation 5).
(4)$${\hat{F}}_{vi}=F_{vi}⊕\left(\left(F_{ir}-F_{vi}\right)⊗AP\left(FC\left(F_{ir}-F_{vi}\right)\right)\right)$$
(5)$${\hat{F}}_{ir}=F_{ir}⊕\left(\left(F_{vi}-F_{ir}\right)⊗AP\left(FC\left(F_{vi}-F_{ir}\right)\right)\right)$$
where AP(·) denotes the average pooling operation, compressing the difference map into a channel descriptor vector by calculating the mean of each channel. FC(·) represents two fully connected layers with ReLU activations, which generate channel weights after the two FC layers. ⊗ refers to matrix multiplication, multiplying the original difference map with the channel weights to obtain an amplified difference map. ⊕ indicates a matrix addition, finally adding the enhanced difference map back to the feature maps.

### 3.5. Loss Function

To facilitate our progressive fusion framework to adaptively integrate meaningful information based on temperature conditions, an innovative temperature color perception loss is designed. Temperature color perception loss $L_{TC}$ is precisely defined as Equation (Equation 6).

Where $L_{int}^{ir}$ and $L_{int}^{vi}$ represent the intensity losses for the infrared and visible images, respectively. $W_{ir}$ and $W_{vi}$ are the temperature-aware chromatic weights. The intensity losses are specifically defined as shown in Equations (Equation 7) and (Equation 8).
(6)$$L_{TC}=W_{ir}\cdot L_{int}^{ir}+W_{vi}\cdot L_{int}^{vi}$$
(7)$$L_{int}^{ir}=\frac{1}{HW}{∥I_{f}-I_{ir}∥}_{1}$$
(8)$$L_{int}^{vi}=\frac{1}{HW}{∥I_{f}-I_{vi}∥}_{1}$$
where *H* and *W* are the height and width of the input images, and ${∥\cdot ∥}_{1}$ denotes the L1 norm. Indeed, the intensity distribution of the fused image ought to retain consistency across various source images corresponding to different temperature heat patterns. Therefore, the multispectral weights $W_{ir}$ and $W_{vi}$ are used to modulate the intensity constraints for the fused image.

Additionally, the aim for the fused image is to simultaneously maintain optimal temperature color distribution and abundant texture details. Therefore, both maximum pixel loss and texture loss are introduced, defined as shown in Equations (Equation 9) and (Equation 10).
(9)$$L_{\max P}=\frac{1}{HW}{∥I_{f}-\max\left(I_{ir},I_{vi}\right)∥}_{1}$$
(10)$$L_{edge}=\frac{1}{HW}{∥E_{f}-E_{ir}-E_{vi}∥}_{1}$$
where max(·) denotes element-wise maximum selection, and *E* denotes a binarized edge map extracted by the Canny operator, which is used to measure the textural information of an image.

## 4. Experiment

### 4.1. Dataset Construction and Experimental Settings

In this section, experiments were conducted to validate the proposed fusion method in two different scenarios: using a publicly available dataset and a self-constructed dataset focused on electric power equipment. After providing detailed descriptions of the experimental settings for both the training and testing phases, multiple ablation studies were performed to investigate how various components of the proposed fusion network influence performance. Subsequently, qualitative comparisons with other existing algorithms were conducted, and multiple performance metrics were utilized to objectively evaluate the effectiveness of the fusion framework. TGLFusion was implemented in a programming environment using PyTorch (created by Facebook AI Research and sourced from Menlo Park, CA, USA) and executed on an NVIDIA 3090Ti GPU (manufactured by NVIDIA Corporation and sourced from Santa Clara, CA, USA).

### 4.2. Settings in Training and Testing Phase

In the first training experiment scenario, TGLFusion was trained using the Infrared and Visible publicly available dataset (MSRS). Ref. [6], which consists of 221 images. The dataset was augmented to 4320 pairs of 128 × 128 images using data augmentation techniques such as random flipping, rotation, scaling, cropping, and brightness adjustment. In the second experimental scenario, the dataset used is a self-constructed dataset consisting of 752 pairs of infrared–visible images of power equipment. TGLFusion was trained for 100 epochs using the Adam optimizer, with each epoch randomly sampling 400 pairs of images from the dataset and a batch size of 4. The learning rate was initialized at 0.001 and gradually reduced to 0.0001 after 80 epochs. In the testing scenario, the test images consisted of 20 image pairs sourced from the Infrared and Visible Light publicly available dataset (TNO) [35] and 50 selected image pairs from our self-constructed electric power equipment dataset. The fusion algorithm was objectively evaluated using six image evaluation metrics. Finally, TGLFusion was compared against other algorithms, including DenseFuse [17], FusionGAN [22], PMGI [5], DDcGAN [36], SeAFusion [37], Deepfuse [38], MDLatLRR [39], GTF [40], and the GANMCC [27] in a comparative experiment.

### 4.3. Tuning of Hyperparameters

In this section, the effectiveness of the threshold segmentation method is first analyzed by adjusting the size of *T* to determine the optimal value, as illustrated in Figure 4. Following that, the impact of adjusting the value of α on the relative weight of infrared and visible features in fusion performance is investigated, as depicted in Figure 5.

Using a threshold-based approach for segmenting the infrared image aims to extract significant regions. Considering the linear relationship between the grayscale values in thermal infrared images and the temperature they represent [41], the value of *T* is initially set within the range of 100–240. The evolution of significant region extraction is studied as the threshold value *T* is progressively varied.

Several sets of infrared images from the MSRS dataset were selected to validate the effect of threshold segmentation under different values of *T*, as depicted in Figure 4. According to the subjective results, a low *T* results in the excessive inclusion of irrelevant background pixel information, while a high *T* leads to a loss of critical temperature information related to the primary thermal targets. Ultimately, the value of *T* was set at 200, as it best reflects the real thermal conditions of the target compared to the original image.

With the *T* value fixed, the optimal high-temperature ratio in the infrared image can then be calculated using the above formula. In the next step, the optimal value of α is determined, weights for the infrared and visible features are calculated, and model training proceeds. The value of α is gradually adjusted, and 4320 pairs of images from the MSRS dataset are used for training, with testing and analysis conducted on the TNO dataset. Some of the subjective fusion results are shown in the figure below. It can be observed that α, as the scaling factor in the formula, largely determines the weight of infrared and visible features. When α is small, the weight of the infrared features is small, and the fusion result tends to favor the visible image, losing the infrared features. Conversely, when α is large, the fusion result incorporates too much infrared information, leading to spectral contamination. Ultimately, the value of α is set to 5, as it best balances the weights of the infrared and visible features and highlights significant targets in the fused image. Additionally, objective results indicate that setting α to 5 yields optimal image quality metrics, as shown in Table 3.

Finally, with the determined hyperparameter values, the weights can be calculated, and partial results are shown in Figure 6. It can be observed that, in infrared images 1, 2, and 3, where there is a higher amount of thermal information, larger weights are calculated after segmentation for the infrared image. In contrast, in infrared images 4, 5, and 6, where there is less thermal information, smaller weights are calculated for the infrared image. Therefore, in image pairs with more temperature information, the fused image will highlight the infrared image more prominently. In image pairs with less temperature information, the fused image will emphasize the visible image more prominently. This aligns with the proposed temperature-guided mechanism module.

### 4.4. Analysis of the Public Dataset

After fixing the values of *T* and α, the model obtained through training is the best-performing fusion model. Subsequently, the fusion performance of the model was further tested using the TNO dataset, employing both subjective observations and objective metric evaluations. Additionally, several mainstream fusion algorithms were selected for comparison. Sets of fusion results on the TNO dataset are shown in the figure below.

#### 4.4.1. Qualitative Comparison

In the first and second scenarios, most methods were unable to effectively preserve the background information from the visible image and the thermal information from the infrared image. Algorithms belonging to GANs such as DDcGAN, FusionGAN, and GANMcC, as they generate images anew instead of employing static or dynamic strategies to fuse multispectral image features, tend to produce fusion results with blurred pixel information.

While DeepFuse, DenseFuse, GTF, mdlatrr, and PMGI retained the edge texture background information from the visible image relatively well, they simply fused the infrared and visible images without considering how to highlight their respective features and significant areas, resulting in the thermal information from the infrared being less pronounced in the fusion results.

Only SeAFusion and our method managed to effectively retain both the edge texture background information from the visible image and the significant thermal information from the infrared image. In Figure 7, within the red box, many fusion methods failed to effectively preserve the significant features of the ’branches’ from the visible image, except for SeAFusion and our proposed method. This suggests that other fusion methods integrated too much background information from the infrared image, or the fusion results resulted in pixel loss compared to the source images.

Compared to all these fusion methods, within the green box, most methods were able to preserve the significant ’face’ features to some extent from the infrared image. However, our method emphasized a clearer face.

In Figure 8, within the red box, except for SeAFusion and our method, the results of other fusion methods still displayed the very blurry features of the ’trunk.’ This suggests that other fusion methods integrated too much background information from the infrared image, or the fusion results resulted in pixel loss compared to the source images.

In the third and fourth scenarios, where the background information is relatively simple and the distribution of thermal information in the infrared is more concentrated, most algorithms struggled to preserve the infrared information while retaining the background information from the visible image.

In Figure 9, within the green box, DeepFuse, DenseFuse, GTF, and MDLatLRR almost failed to preserve the infrared information of the ’person,’ while the results of GAN algorithms still showed blurred distortions in the ’person.’ Only PMGI, SeAFusion, and our algorithm managed to preserve the ’person’.

In Figure 10, the results similar to those in Figure 3 were observed. Only PMGI, SeAFusion, and our algorithm could preserve the ’person.’ However, in the green and red boxes, our algorithm was the most capable of retaining the background texture information of the ’chimney’ and the ’person.’ Compared to all these fusion methods, DeepFuse, DenseFuse, GTF, and MDLatLRR struggled to effectively fuse the infrared thermal information and could only retain some target edge information from the infrared image. As previously mentioned, GAN-based algorithms like DDcGAN, FusionGAN, and GANMcC produced distorted fusion results, making them appear more blurry and leaning toward or even accentuating the infrared image. They lost a significant amount of visible information. Only PMGI, SeAFusion, and our method effectively retained both the visible and infrared information. However, our algorithm had overall higher brightness, maximally restoring the lighting information from the visible image and emphasizing the significant areas in the infrared image, enhancing the overall contrast.

#### 4.4.2. Quantitative Comparison

To further validate the effectiveness of the proposed method, four evaluation metrics, including AG, SF, PSNR, and MI, were utilized. All metric testing experiments were conducted on the TNO dataset. The results of the four objective evaluation metrics on TNO are shown in Table 4. It can be observed that, except for the AG metric, our method achieved the highest average values across all metrics. In terms of the AG metric, our method was slightly inferior to DDcGAN. Notably, our method attained the maximum value for the SF metric, indicating superior performance in preserving spatial frequency information, with the fused images retaining more texture and structural details. The highest PSNR value signifies a high peak signal-to-noise ratio between the fused and original images, implying minimal differences between them. The highest MI value indicates that our method successfully conveyed more features from the source images to the fused images. As depicted in Figure 11, our method exhibited the best average distribution across SF, PSNR, and MI metrics.

### 4.5. Analysis on the Electric Power Equipment Image Dataset

The self-constructed dataset of electric power equipment images is shown in Figure 12. This dataset comprises 752 pairs of infrared–visible images of electric power equipment, registered using a contour angle directional method [41]. These images were captured using FLIR thermal infrared sensors. Among these pairs, 441 images belong to high-temperature equipment, indicating the presence of higher temperature information and significant areas in the infrared images, while 311 images are associated with low-temperature equipment, representing less temperature information and significant areas in the infrared images. The image resolution is 640 × 480. During the fusion process, the infrared images were first transformed into the YCbCr color space. Then, the fusion model combined the Y channel of the infrared image with the visible image to create the fused image. Finally, the fused image was transformed into the RGB color space by combining it with the Cb and Cr channels from the infrared image [6].

#### 4.5.1. Qualitative Comparison

To visually demonstrate the adaptability of our temperature-based color perception fusion method to temperature variations, three sets of high-temperature and low-temperature electric power equipment visible and infrared image pairs, along with their fusion results, were selected for subjective visual analysis and comparison, as depicted in Figure 13.

In high-temperature scenarios, inspectors primarily focused on the heat generation of electric power equipment. The thermal radiation information from the infrared image should be highlighted as a significant element in the fusion result, while the realistic texture details from the visible image serve as supplementary information. Therefore, an excellent fusion algorithm should retain the texture details of the visible image while emphasizing the significant targets without introducing spectral contamination. In the high-temperature electric power equipment scenes numbered 1, 2, and 3 in the above figure, FusionGAN and GTF effectively highlighted the temperature information while preserving the details of the visible image. However, the fusion images became blurred with reduced clarity. Except for SeAFusion and our algorithm, other algorithms fused non-essential color information from the visible image, leading to a reduction in the temperature information from the infrared, causing spectral contamination. Dark spots appeared in the fusion images where the corresponding areas in the infrared image were extremely bright (indicating high temperatures), with DDcGAN, GANMcC, and MDLatLRR showing more severe issues. Our model, on the other hand, maximally preserved and emphasized critical temperature information in high-temperature scenarios, without losing the fine texture details from the visible image.

In low-temperature scenarios, the thermal radiation information in the infrared image contains limited temperature information and is not as critical. In such situations, the rich information in the visible image can complement the deficiencies in the infrared image. Inspectors may focus on aspects such as equipment appearance, potential damage, and environmental stability in low-temperature working environments. Therefore, fusion images in these scenarios are more inclined to retain the realistic texture details from the visible image. In the low-temperature electric power equipment scenes numbered 4, 5, and 6 in the figure above, it can be observed that, except for SeAFusion and our algorithm, other algorithms overly fused the temperature colors from the infrared image, resulting in lower brightness and contrast in the low-temperature infrared images. This caused the overall fusion images to appear dark and led to the loss of the bright information from the visible image. Such fusion images would be challenging to observe and analyze. FusionGAN, while maintaining higher brightness and contrast, introduced other information, causing significant and inappropriate changes in the temperature colors of the electric power equipment in the fusion image compared to the source image, leading to a loss of accuracy and objectivity. In contrast, our model excels in low-temperature thermal radiation scenarios by maintaining higher brightness and contrast in the overall fusion image and preserving the realistic texture details from the visible image. It compensates for the shortcomings of weak information in low-temperature infrared images.

In practical application scenarios such as electric power equipment inspection, the imbalance in information between infrared and visible images can make it challenging for the most simple and direct fusion methods to highlight the characteristics of different spectra according to actual requirements. The introduction of a loss function based on temperature perception and the implementation of a multispectral information complement module have effectively addressed this issue. Our model can adaptively adjust the spectral information weights based on the temperature distribution, enabling infrared and visible light to complement each other effectively in the fusion image. This capability is particularly valuable in real-world scenarios where the fusion of these two types of images is essential for various inspection and analysis tasks.

#### 4.5.2. Quantitative Comparison

Table 5 presents the comparative results of four fusion image evaluation metrics, SF, AG, MI, and PSNR, for 50 pairs of high-quality electric power equipment infrared–visible image pairs. In the table, red indicates the best values, and blue indicates the second-best values. It is noteworthy that our method exhibits a significant advantage in three of the metrics. The highest SF and second-highest AG values suggest that the fused images have the best preservation of edges and textures and appear sharper and clearer visually. The highest MI and PSNR values indicate a high level of consistency between the fused image and the source image, with lower distortion. This indicates that the fusion algorithm can effectively transfer the most spectral information from the source image to the fused image based on temperature conditions, reducing information loss and maintaining the quality and details of the source image. Furthermore, as shown in Figure 14, our method consistently demonstrates the best performance across all metrics, reflecting its superiority in various aspects of fusion algorithm performance.

Table 6 presents the measurements of parameters, model size, and FLOPs obtained by the implemented fusion model based on deep learning in the aforementioned image fusion method with the given input size of (1, 1, 640, 480). These measurements serve to evaluate the model’s volume and lightweight characteristics. In the table, values highlighted in red indicate the optimal values, while those in red represent the next best values. It can be observed that the fusion model implemented with MobileBlock demonstrates the optimal parameter quantity and model size. This signifies that MobileBlock significantly achieves a lightweight design compared to existing fusion models, enhancing the feasibility of running on edge devices.

### 4.6. Ablation Experiment

To demonstrate the effectiveness of the proposed temperature-color-perception loss function, MICM, and MobileBlock, other model variants were constructed and evaluated, including those where the temperature-color-perception loss function, MICM, or MobileBlock was individually removed. The removal was achieved by setting the corresponding factors to 1, making the loss function ineffective. The method of removing MICM involved directly transmitting between layers without additional computation. Removing MobileBlock replaced the convolutional structure block with a simple 3×3 standard convolution.

Table 7 and Figure 15 shows that, after removing the temperature-color-perception loss function, all metrics significantly decreased. The fused image lost more information, pixel values were more concentrated, contrast was lower, differences between the fused image and the original image increased, and image sharpness and fidelity decreased. This indicates that our temperature-color-perception loss function effectively adapts the contributions of infrared and visible light by preserving more information-rich parts.

After removing the multispectral information complement module, MI and PSNR decreased more, indicating significant distortion and loss in the fused image compared to the original image. This suggests that our multispectral information complement module effectively supplements information differences in various spectral features, enabling the model to integrate various spectral information during the feature extraction stage, ensuring information transfer performance throughout the network.

When MobileBlock was removed, not only did the image quality metrics decline, but the model parameters, size, and computational load all increased by more than twice. This demonstrates that MobileBlock, as a transmission layer, provides reliable information transfer and lightweight performance.

## 5. Discussion

Due to the distinct physical properties and imaging mechanisms of these two modalities, infrared images convey temperature information on the targets, while visible images provide higher spatial and textural information [3]. This information imbalance affects the interpretation of fused images differently across various usage scenarios [42]. Previous work has addressed the issue of unbalanced spectral weight allocation by considering illumination conditions [6], but temperature conditions still need to be taken into account. For instance, in scenarios such as high-temperature electric power equipment, prioritizing the display of the equipment’s thermal conditions in the fused image is preferred. Conversely, in low-temperature environments, emphasizing the visual appearance of the equipment in the fused image is more desirable. Therefore, appropriately allocating weights to the imbalanced spectral information is necessary.

Existing fusion methods typically achieve fusion by appropriately combining infrared and visible images. These methods include but are not limited to pixel-level, feature-level, and model-level fusion approaches [43]. In pixel-level fusion [44], the pixel values of the two images are usually combined through simple weighted averaging or by following certain rules such as minimum–maximum rules or mean rules. In feature-level fusion [45], local or global features of the images are typically utilized for fusion. In model-level fusion [46], deep learning models are commonly employed to learn the mapping relationship between the two images, enabling more sophisticated fusion. However, these methods often do not consider how to handle spectral information and instead directly adopt simple feature fusion rules, which may not effectively reflect the information of the two images in different scenarios.

The primary focus of TGLFusion is to address the issue of information imbalance between infrared and visible images, as well as the computational complexity of the image fusion model. This is achieved by designing an temperature-guided mechanism module that dynamically adjusts the weights of different spectral information during the fusion process based on the temperature of the infrared image. This ensures a better representation of the features from both types of images. Additionally, the proposed method introduces MICM and MobileBlock, enhancing the model’s capability to handle information imbalance while significantly reducing computational complexity.

In the experimental evaluation phase, TGLFusion was compared with nine other fusion methods using both subjective and objective approaches. In terms of subjective comparison, the results show that, in the four scenes of the TNO dataset, the fused images generated by TGLFusion preserve more realistic temperature information and clearer texture information compared to most methods. In the electrical equipment scene, TGLFusion flexibly highlights infrared temperature information in high-temperature scenes and emphasizes visible texture information in low-temperature scenes, outperforming the other methods. Regarding objective comparison, on the TNO dataset, TGLFusion achieved optimal values in three out of four image fusion evaluation metrics—AG, SF, PSNR, MI. In the electrical equipment scene, TGLFusion attained optimal values in all metrics. TGLFusion not only outperformed other models in image fusion effects but also, compared to other deep learning methods, achieved optimal values in lightweighting metrics.

## 6. Conclusions

In this study, a lightweight fusion method called TGLFusion for infrared and visible images is proposed, which utilizes temperature information from significant regions in infrared images to compute multi-spectral weights. The aim of this method is to address the issues of spectral information imbalance and high computational complexity in image fusion models. A temperature-guided mechanism module is designed to estimate the infrared temperature information and allocate spectral weights accordingly. Additionally, a temperature color perception loss is constructed based on these spectral weights to guide the model training. The fusion model is implemented using the designed MICM and MobileBlock, ensuring both complementary information transfer and significant model lightweighting. Comprehensive comparative experiments were conducted on the TNO and electric power equipment datasets, comparing TGLFusion with nine other advanced algorithms. The results demonstrate that TGLFusion outperforms other methods in various scenarios, effectively highlighting the important information from both infrared and visible images, and achieves the best model lightweighting. The fused images generated can be more effectively used for other advanced computer vision tasks. Furthermore, practical experiments on the electric power equipment datasets demonstrate the feasibility of TGLFusion in a specific industry.

## Figures and Tables

**Figure 1 sensors-24-01735-f001:**
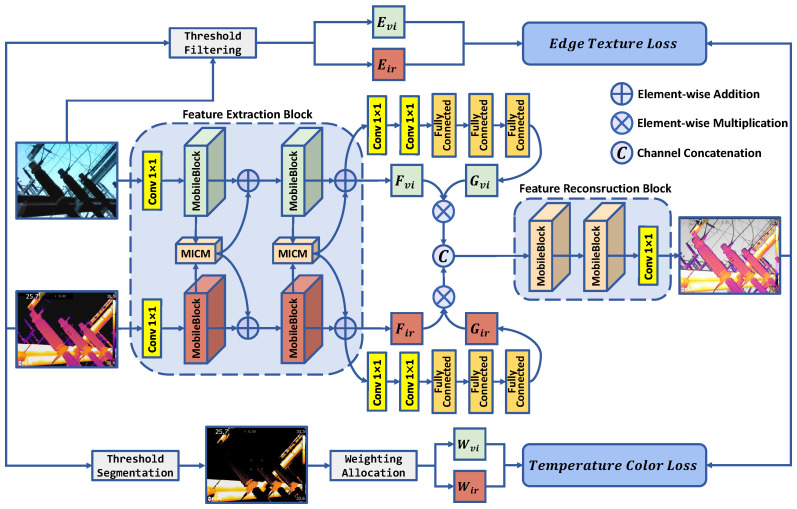
Model structure.

**Figure 2 sensors-24-01735-f002:**
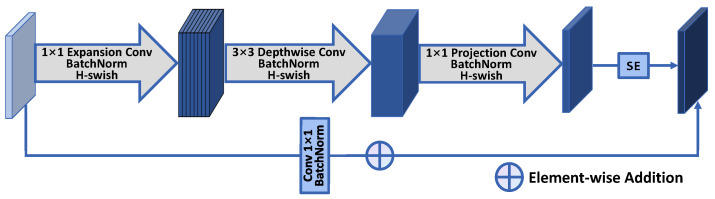
MobileBlock.

**Figure 3 sensors-24-01735-f003:**
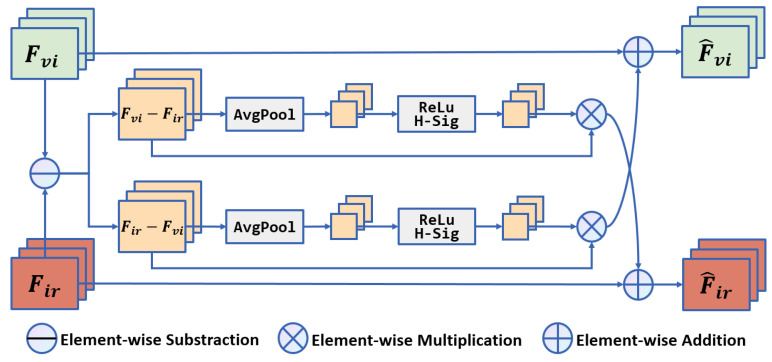
Multispectral information complementary module.

**Figure 4 sensors-24-01735-f004:**
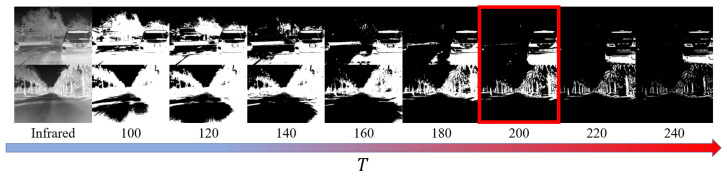
The effect of threshold segmentation in the infrared image under different values of *T*, and the red box indicates the best visual performance.

**Figure 5 sensors-24-01735-f005:**
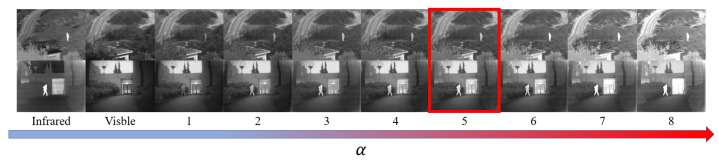
The fusion results under different values of α, and the red box indicates the best fusion result.

**Figure 6 sensors-24-01735-f006:**
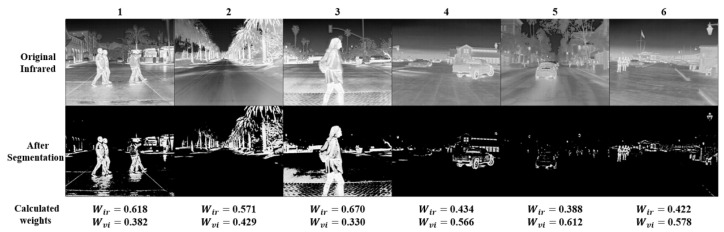
The calculated weight values, with more infrared information in images 1, 2, and 3, and less in images 4, 5, and 6.

**Figure 7 sensors-24-01735-f007:**
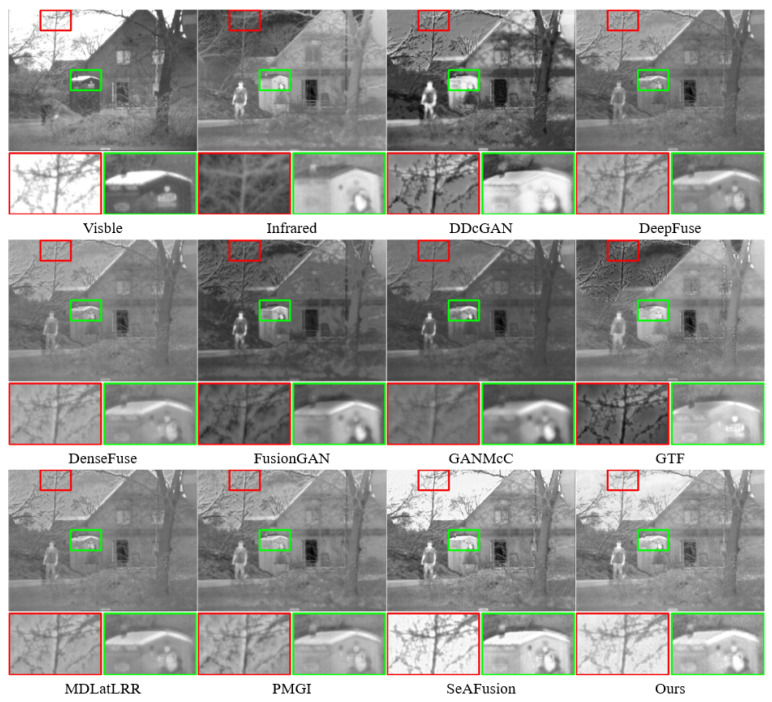
Qualitative comparison of the proposed method with SOTA methods on 2 men in front of a house. For a more distinct comparison, a texture-rich area and a prominently targeted area (i.e., green box and red box) are outlined in each image.

**Figure 8 sensors-24-01735-f008:**
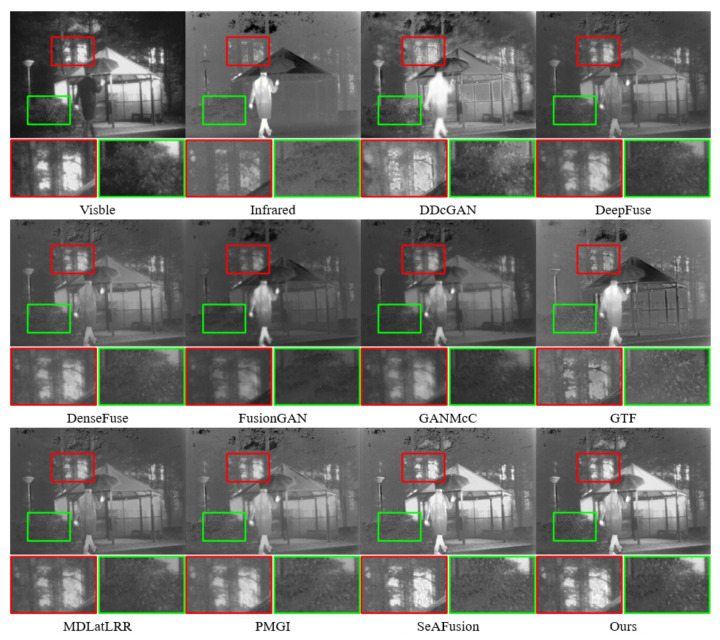
Qualitative comparison of the proposed method with nine SOTA methods on Kaptein 1654. For a more distinct comparison, a texture-rich area and a prominently targeted area (i.e., green box and red box) are outlined in each image.

**Figure 9 sensors-24-01735-f009:**
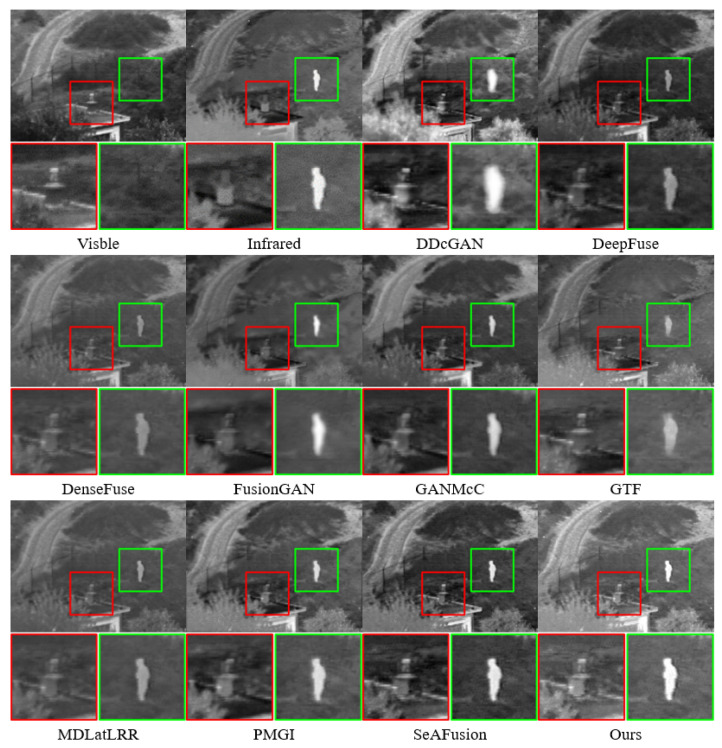
Qualitative comparison of the proposed method with nine SOTA methods on Nato camp sequence. For a more distinct comparison, a texture-rich area and a prominently targeted area (i.e., green box and red box) are outlined in each image.

**Figure 10 sensors-24-01735-f010:**
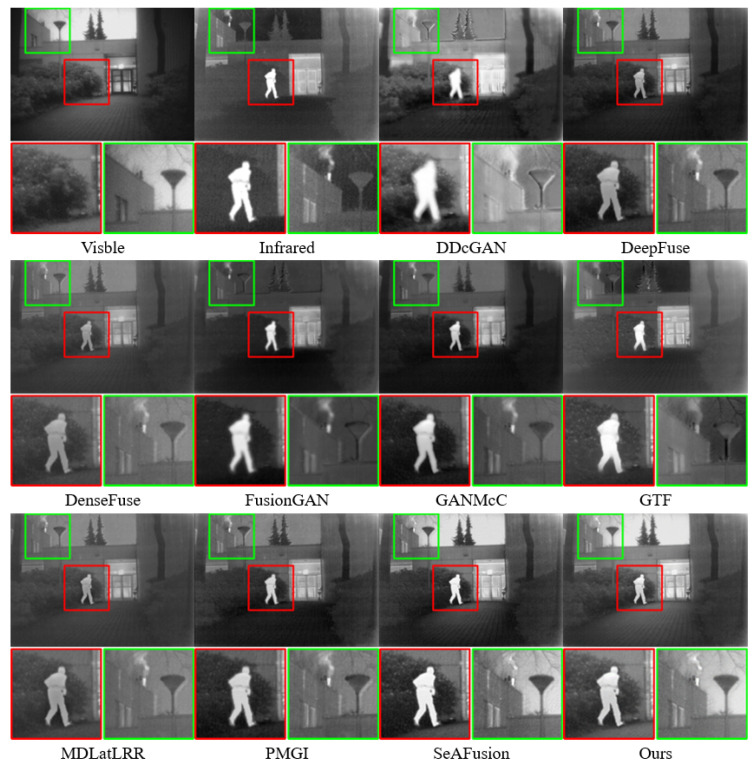
Qualitative comparison of the proposed method with nine SOTA methods on Kaptein 1123. For a more distinct comparison, a texture-rich area and a prominently targeted area (i.e., green box and red box) are outlined in each image.

**Figure 11 sensors-24-01735-f011:**
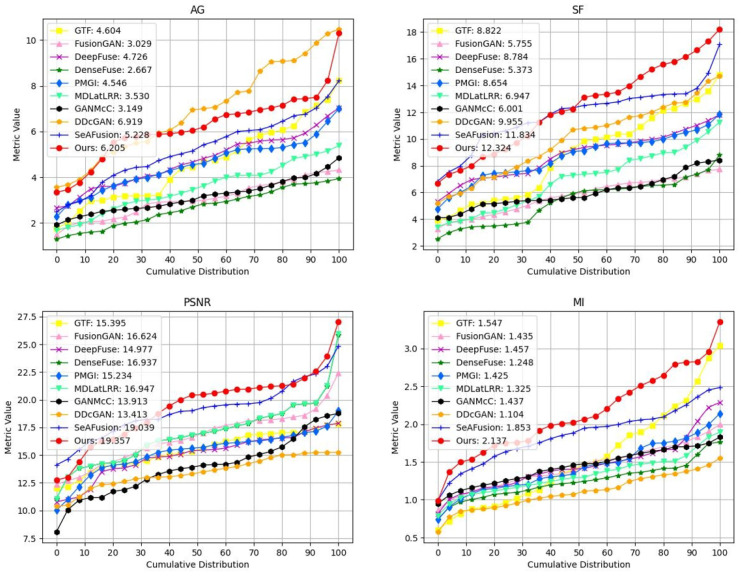
Quantitative comparisons of the four metrics on 20 image pairs from the TNO dataset. A point (x,y) on the curve denotes that x% of image pairs have metric values that are no more than *y*.

**Figure 12 sensors-24-01735-f012:**
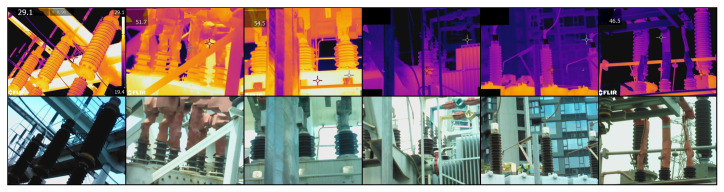
Partial pairs of electric power equipment images.

**Figure 13 sensors-24-01735-f013:**
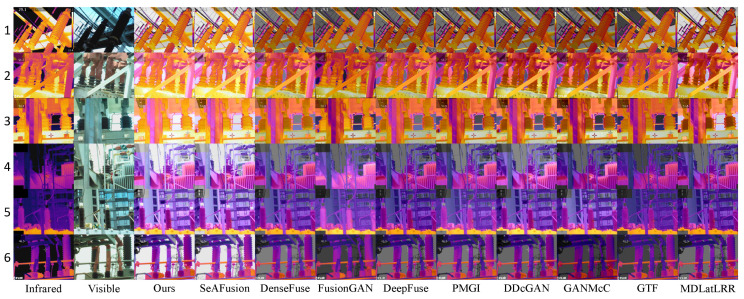
In high-temperature and low-temperature electric power equipment scenarios, the proposed method was subjectively compared with nine other methods.

**Figure 14 sensors-24-01735-f014:**
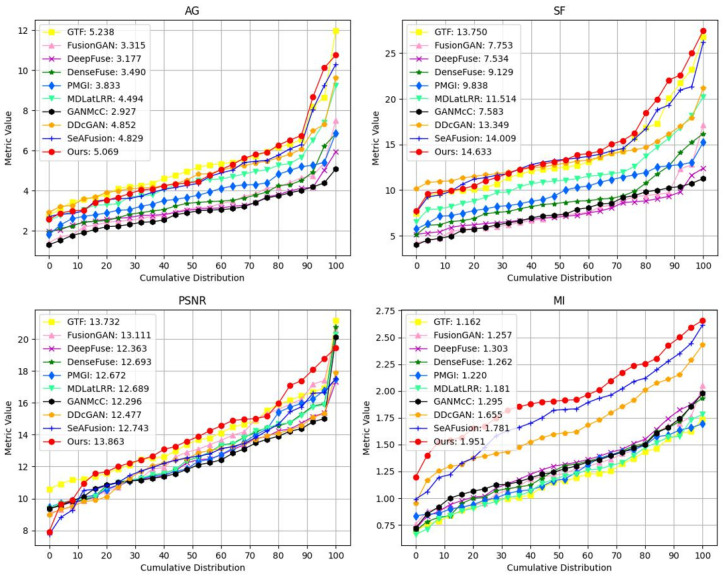
Quantitative comparisons of the four metrics on 50 image pairs from the electric power equipment dataset. A point (x,y) on the curve denotes that x% of image pairs have metric values that are no more than *y*.

**Figure 15 sensors-24-01735-f015:**
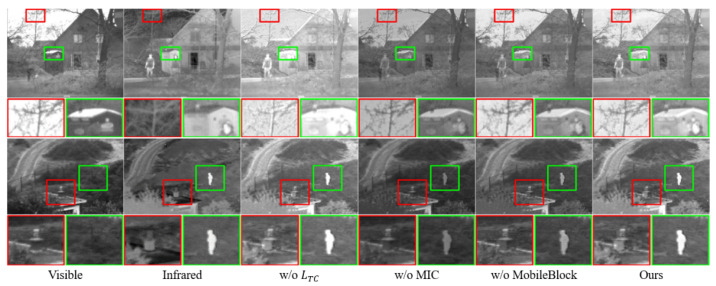
Ablation experiment. For a more distinct comparison, a texture-rich area and a prominently targeted area (i.e., green box and red box) are outlined in each image.

**Table 1 sensors-24-01735-t001:** Feature extractor and image reconstructor blocks.

	Feature Extractor Block			Image Reconstructor Block		
	Layer	Input Channels	Output Channels	Layer	Input Channels	Output Channels
Layer1	Conv 1 × 1	1	16	MobileBlock	128	64
Layer2	MobileBlock	16	32	MobileBlock	64	32
Layer3	MobileBlock	32	64	Conv 1 × 1	32	1

**Table 2 sensors-24-01735-t002:** Layer Operations.

Input	Layer	Operator	Output
c×h×w	Expansion Conv	1 × 1 conv2d, BN, H-swish	(tc)×h×w
(tc)×h×w	Depthwise Conv	3 × 3 conv2d, BN, H-swish	(tc)×h×w
(tc)×h×w	Projection Conv	1 × 1 conv2d, BN, H-swish	c×h×w

**Table 3 sensors-24-01735-t003:** The average metrics values of the TGLFusion with different α on 20 images from TNO.

α	AG	SF	PSNR	MI
1	5.956	11.397	15.495	1.760
2	6.194	13.210	16.833	1.925
3	6.151	12.185	18.821	2.054
4	6.118	12.173	18.743	2.091
5	6.205	12.324	19.357	2.137

**Table 4 sensors-24-01735-t004:** The quantitative results on 20 image pairs from TNO, the best results are highlighted in red, and the second best results are highlighted in blue.

Method	AG	SF	PSNR	MI
GTF	4.604	8.822	15.395	1.547
FusionGAN	3.029	5.755	16.624	1.435
DeepFuse	4.726	8.784	14.977	1.457
DenseFuse	2.667	5.373	16.937	1.248
PMGI	4.546	8.654	15.234	1.425
MDLatLRR	3.530	6.947	16.947	1.325
GANMcC	3.149	6.001	13.913	1.437
DDcGAN	6.919	9.955	13.413	1.104
SeAFusion	5.228	11.834	19.039	1.853
Proposed model	6.205	12.324	19.357	2.137

**Table 5 sensors-24-01735-t005:** The average image metrics calculated using 50 pairs of electric power equipment infrared and visible images, the best results are highlighted in red, and the second best results are highlighted in blue.

Method	AG	SF	PSNR	MI
GTF	5.238	13.750	13.732	1.162
FusionGAN	3.315	7.753	13.111	1.257
DeepFuse	3.177	7.534	12.363	1.303
DenseFuse	3.490	9.129	12.693	1.262
PMGI	3.833	9.838	12.672	1.220
MDLatLRR	4.494	11.514	12.689	1.181
GANMcC	2.927	7.583	12.296	1.295
DDcGAN	4.852	13.349	12.477	1.655
SeAFusion	4.829	14.009	12.743	1.781
Proposed model	5.069	14.633	13.863	1.951

**Table 6 sensors-24-01735-t006:** Model lightweight metrics measured with an input size of (1, 1, 640, 480), the best results are highlighted in red, and the second best results are highlighted in blue.

Model	Parameters	Model Size (MB)	FLOPs (G)
FusionGAN	924,673	3.698	551.006
DeepFuse	114,497	0.457	70.257
DenseFuse	74,193	0.296	45.475
GANMcC	1,862,209	7.448	1109.800
DDcGAN	212,721	0.850	130.498
SeAFusion	166,657	0.667	101.931
Proposed model	57,313	0.229	34.573

**Table 7 sensors-24-01735-t007:** The contributions of different modules to fusion performance.

	AG	SF	PSNR	MI	Parameters	Size (MB)	FLOPs (G)
Proposed model	5.029	12.805	65.020	2.770	57,313	0.229	34.573
w/o $\;L_{TC}$	4.863	12.644	64.841	2.507	—	—	—
w/o MICM	4.829	12.543	63.219	2.514	—	—	—
w/o MobileBlock	3.863	11.246	59.512	2.096	143,265	0.586	87.805

## Data Availability

The raw data supporting the conclusions of this article will be made available by the authors upon reasonable request.

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
