# Peer review of "TGLFusion: A Temperature-Guided Lightweight Fusion Method for Infrared and Visible Images"

_sensors, 2024, doi:10.3390/s24061735_

Round 1

Reviewer 1 Report

Comments and Suggestions for Authors

1. Kindly avoid the term 'we' in the manuscript.

2. Kindly avoid the group citation.

3. Why is the fusion of infrared and visible images essential in computer vision, and what challenges does it address in real-world applications?

4. What is the common issue related to information imbalance between infrared and visible images, and how does it impact the interpretation of the fused images?

5. How do existing fusion methods typically handle the fusion of infrared and visible images, and what limitations do they exhibit, especially concerning temperature and edge texture information?

6. What is the primary focus of the proposed method, and how does it address the issue of information imbalance between infrared and visible images?

7. Explain the concept of distribution proportion of infrared pixel values and how it contributes to the adaptive weight allocation in the proposed method?

8.What role does the weight allocation mechanism play in the proposed method, and how does it enhance the fusion capabilities of the model?

9. How is the MobileBlock with a multispectral information complementary module integrated into the proposed method, and what advantages does it offer in terms of model lightweightness and information compensation?

10. Provide insights into the training process of the proposed method, particularly regarding the temperature-color-perception loss function and its role in adaptive weight allocation?

11. What are the key findings and results from the experimental evaluation of the proposed method, especially in scenarios such as the electric power equipment scene and publicly available datasets?

12. Add a separate section for Discussion and brief the research findings with the literature support.

13. How does the proposed method compare to mainstream fusion methods, and what specific areas of improvement or superiority does it demonstrate?

14. In what ways does the proposed method contribute to the field of computer vision, especially in scenarios that require the fusion of infrared and visible images on edge devices for real-time processing?

15. Revise the conclusion with research findings.

Comments on the Quality of English Language

Moderate editing of the English language is required

Reviewer 2 Report

Comments and Suggestions for Authors

The paper “TGLFusion: A Temperature-Guided Lightweight Fusion Method for Infrared and Visible Images” introduces an approach to thermal and vision image fusion. The paper is well organized and contains sufficient literature review, but contains the following significant limitations, which include methodological problems.

Limitation:

1. Line 147. 'This is an example of a quote.' Seems like text from a template.

2. Line 155. The statement “Usually, in the infrared image, the areas with higher temperatures also have larger grayscale values.” Is not clear. Do you mean that pixels with higher temperature have higher value (i.e. a pixel value in direct ratio with temperature)?

3. Equation 1. The expression inside the sum does not have an index of summation. Also, it is the sum over all pixels, and there is no defined value of pixels with value lower than T.

4. Equation 2. The r (in a power of exp) is not defined value.

5. Line 176. It is not clear why the values were chosen as chosen. Provide a justification. According to the introduction of the paper, this value can be related to the type of landscape in an image.

6. Lines 280-282, The size of the test subset is not defined, and also, it is not clear how and how many images were chosen from the 'self-constructed power equipment dataset'.

7. Lines 296-301. There are no clear formal criteria for optimization of the T value.

8. Lines 302-313. There are no clear formal criteria for optimization of the alpha value.

9. Section 4.5. Was the proposed model trained on “electric power equipment Image Dataset” and compared with other approaches trained on other datasets? This approach introduced bias.  You should compare networks trained on the same datasets (train all models on the custom dataset) or test on unseen datasets for all models (use the model trained in Section 4.4).

10. Figs. 10 and 13. It is not clear how this distribution was estimated. The CDF is equal to 1 approaching infinity by definition.

11. Finally, the research is difficult to reproduce because of the absence of publicity available source code.

Therefore, the article requires a significant revision and cannot be accepted for publication.

Reviewer 3 Report

Comments and Suggestions for Authors

1.      How does the TGLFusion method account for varying illumination conditions in visible images, and is there a mechanism to normalize illumination before the fusion process to ensure consistent temperature-guided weight allocation? The article does not explicitly address the normalization of illumination conditions in visible images before fusion. However, it's important to note that the focus of TGLFusion is on leveraging temperature information from infrared images and texture information from visible images. Incorporating an illumination normalization step could enhance the model's robustness by ensuring consistent temperature-guided weight allocation under varying lighting conditions. This adjustment could be a valuable addition to the method's preprocessing stage. 

2.      The article showcases the efficiency of TGLFusion's design, particularly with the integration of MobileBlock, which notably cuts down computational demands. Yet, it falls short of offering concrete benchmarks regarding its real-time processing performance on edge devices. Considering the importance of real-time processing for applications such as surveillance, autonomous driving, and various edge computing tasks, providing a detailed comparison of processing speeds and resource usage on real hardware could greatly bolster the paper's relevance and applicability in real-world scenarios. This assessment is crucial for understanding its applicability and efficiency in scenarios where immediate data analysis is essential and computing power is at a premium.

3.      How does the TGLFusion method handle the variation in temperature scales across different infrared cameras, and is there a calibration process to standardize the input before fusion? The manuscript lacks specific details on handling variations in temperature scales across different infrared cameras, which is crucial for consistency in fused images. Standardizing the input before fusion, possibly through a calibration process that aligns temperature readings to a common scale, could improve the method's adaptability to various infrared imaging systems. Addressing this could help in broadening the application range of TGLFusion.

4.      The article introduces a novel temperature-color-perception loss function but does not compare its performance with traditional loss functions regarding convergence speed and training stability. How does the proposed temperature-color-perception loss function compare with traditional loss functions in deep learning-based image fusion in terms of convergence speed and stability during training? Such a comparison could provide valuable insights into the efficiency and effectiveness of the proposed loss function, helping to validate its contribution to the field of deep learning-based image fusion.

5.      The document does not mention how TGLFusion addresses potential misalignment issues between infrared and visible images due to the different field of views or perspectives of the sensors, which is critical for accurate fusion. Incorporating an alignment mechanism or preprocessing step to correct for any discrepancies in sensor perspectives or fields of view before fusion would be essential for ensuring the quality of the fused image.

6.      How scalable is the TGLFusion approach when applied to large-scale datasets? what are the implications for training time and memory requirements? It's vital to know how scalable this method is for it to be used effectively in situations involving large amounts of data, where being able to process data efficiently and manage memory wisely are key.

7.      While the article showcases TGLFusion's application in electric power equipment contexts, it leaves out detailed information on how the method precisely captures complex thermal and structural attributes. Addressing this oversight involves delving into TGLFusion's ability to discern and accurately represent different heat sources, which would illuminate its effectiveness in portraying the nuanced thermal behaviors and structural complexities inherent in such environments.

8.      When we consider about using TGLFusion in medical diagnostics applications, how does it could account the varying thermal properties of different tissues and the influence of temperature on infrared imaging of the human body? Could it be effectively applied for biomedical imaging?

9.      Considering the deployment of TGLFusion for monitoring and diagnostics in the energy sector, how does the method handle the high dynamic range and rapid temperature changes in electrical equipment, ensuring accurate fault detection and thermal mapping?

10.   What evidence do you provide to support the claim of superiority over mainstream fusion methods, specifically in terms of quantitative metrics and independent validations across diverse datasets?

Round 2

Reviewer 1 Report

Comments and Suggestions for Authors

Congrats to the authors.

Kindly remove all 'we' in the manuscript.

In Table 6 kindly change 'ours' to 'Proposed  model'

Comments on the Quality of English Language

 Minor editing of the English language is required

Author Response

Thank you very much for your effort on this manuscript!
The manuscript has now been edited to remove all instances of 'we',you can review these sentences at line 46, 62, 153, 179, 181, 219, 249, 283, 404, 558, 562, 567, Fig 7, 8, 9, 10, Table 5.
Now, all instances of 'ours' in all tables have been replaced with 'Proposed model'.

Reviewer 2 Report

Comments and Suggestions for Authors

Thank you for the manuscript revision. All the mentioned limitations have been fixed. And now the paper can be recommended for publication.

Reviewer 3 Report

Comments and Suggestions for Authors

I'm pleased to let you know that after a careful review, your manuscript is now in excellent condition for publication. All of the earlier concerns have been successfully addressed by the revisions, greatly improving the paper's overall quality and clarity. Your research's breadth and depth are noteworthy for their value and depth, contributing notably to the field.

I appreciate your commitment to polishing the manuscript. I do not doubt that the academic community will value your work and that it will significantly advance current discussions and research in your area.

Author Response

Thank you very much for your effort on this manuscript!
Your suggestions have inspired us significantly, and we will conduct further research in our future work!